# Evaluation of wound healing and anti-inflammatory activity of hydro-alcoholic extract and solvent fractions of the leaves of *Clerodendrum myricoides* (Lamiaceae) in mice

**Alemante Tafese Beyna**[1]*, **Assefa Kebad Mengesha**[1], **Ermias Teklehaimanot Yefter**[2], **Wubayehu Kahaliw**[1]

**1** Department of Pharmacology, School of Pharmacy, College of Medicine and Health Sciences, University of Gondar, Gondar, Ethiopia, **2** Department of Pathology, School of Medicine, College of Medicine and Health Sciences, University of Gondar, Gondar, Ethiopia

* alemante45@gmail.com, Alemante.Tafese@uog.edu.et

## Abstract

### Background

Wounds significantly affect people's quality of life and the clinical and financial burden of healthcare systems around the world. Many of the current drugs used to treat wounds have problems such as; allergies and drug resistance. Hence, the exploration of new therapeutic agents from natural origin may avert this problem. *Clerodendrum myricoides* have long been used to treat wounds in Ethiopia. Despite this, nothing has so far been reported about the wound healing and anti-inflammatory activity of *C. myricoides*. This study aimed to evaluate the wound healing and anti-inflammatory activity of 80% methanol extract and solvent fractions of *C. myricoides* leaves in mice.

### Methods

Leaves of *C. myricoides* were extracted using the maceration technique. The extract was formulated as 5% and 10% w/w ointments. The wound healing activity of the extract was evaluated using excision, incision, and burn wound models whereas the healing activities of solvent fractions were evaluated using the excision wound model. A carrageenan-induced paw edema model was used for the anti-inflammatory test.

### Results

In the dermal toxicity test, 2000 mg/kg of 10% extract was found to be safe. In excision and burn wound models, treatment with 10% and 5% extract showed a significant (p<0.001) wound contraction. Solvent fractions of the extract significantly reduced wound contraction. A significant reduction in periods of epithelialization and favorable histopathology changes were shown by extract ointments. In incision wounds, 10% (p<0.001) and 5% (p<0.01) extracts significantly increase skin-breaking strength. After one hour of treatment, 400 mg/kg (p<0.001) and 200 mg/kg (p<0.05) showed significant reduction in paw edema.

**Data Availability Statement:** All relevant data are within the paper and its supporting information files.

**Funding:** The author(s) received no specific funding for this work.

**Competing interests:** the authors have declared that no competing interests exist.

## Conclusion

Results of this study indicate that 80% methanol extract and the solvent fraction of the leaves of *C. myricoides* possess wound-healing and anti-inflammatory activity and support traditional claims.

## 1. Background

Wound is damage or disruption to the normal anatomical structure, function, and integrity of living tissue [1], and it arises from chemical, thermal, microbial, or physical damage to a tissue or be the result of a disease process [2]. The healing of wounds is a complex and dynamic process of restoring the structure and function of damaged tissues [3].

Wounds, especially chronic wounds, have considerable humanistic and economic burdens [4]. Chronic wounds represent a substantial financial and humanistic challenge for society. They not only diminish the quality of life for individuals grappling with them but also escalate healthcare expenses due to reduced productivity [5]. According to current estimates, 1–2% of people in developed countries will encounter a chronic wound at some point in their lives [6]. Despite wound care being a worldwide issue valued in the multibillion-dollar range, in the US alone, it impacts approximately 5.7 million individuals, accounting for roughly 2% of the population, and incurs an annual expenditure of US$20 billion [4] Chronic wounds cost 1% to 2% of the yearly health care budget in European countries [7]. In South Asian and sub-Saharan African countries, 1% to 2%of people have experienced a chronic wound at some point in their lives [8]. The burden of wound is high in Ethiopia as reported in a study conducted in 2015 in Amhara Regional State Referral Hospitals, In this study, injury was found to 55.6% in visiting emergency departments [9].

Plant-based treatment is widely practiced in Ethiopia to facilitate the healing of these injuries. [10]. In addition, medicinal plants are used for a number of conditions and are widely accepted across various cultures and socio-economic levels [11].

In Ethiopia, *Clerodendrum myricoides*, "Misirich" (Amharic) [12] is used to treat wounds and burns (dried leaves are grounded, powdered and applied on infected part) [13–15], malaria [16], epilepsy and anthrax [17], conjunctiva, and trachoma [12]. Previous reports indicated that *C. myricoides* showed *in vitro* antibacterial [18, 19], and antioxidant activity [20]. Studies in animals reported that, *C. myricoides* showed strong activities against *P. berghei* (antimalarial) [21], antidiarrheal [22], and diuretic effects [23]. Though there is strong ethnobotanical evidence, there hasn't been any scientific research on the plant's ability to manage wounds. Therefore, the purpose of this study was to provide scientific proof supporting the plant's traditional use in wound healing.

## 2. Materials and methods

### 2.1 Drugs, chemicals, and instruments

Drugs, chemicals, and instruments used in this study are hard paraffin (Lab tech chemicals), white soft paraffin (Lab tech chemicals), wool fat (Lab tech chemicals), cetostearyl alcohol (Lab tech chemicals), bees wax (Bo international, India), methanol absolute (Taflen industry), ethyl acetate (Blulux laboratories, India), n-hexane (Blulux laboratories, India), formalin (Sigma diagnostics), hematoxylin (Alpha chemical), eosin (Alpha chemical), tween 80 (Uni-chem, India), distilled water (UoG, medical laboratories), nitrofurazone ointment 0.2% w/v

(Shanghai General Pharmaceutical, China), indomethacin 25 mg (EPHARM, Ethiopia), diazepam 10mg/ml injection (Gland pharm limited, India), ketamine injection (Rotexmedica, Germany), Carrageenan (Triveni Interchem Pvt. Ltd, India).

In addition, hot air oven (Abron, India), weighing balance (Abron, India), lyophilize (Lab freeze, Germany), whatman filter paper (No.1) (Maid stone), rotary evaporator (Yamato, Japan), plethysmometer (Orchid, India), and water bath (Yamato, Japan) were used.

## 2.2 Plant material

The leaves of *C. myricoides* were collected in spring season of Ethiopian November 2022 from Azezo, kebele19, Michael church in Gondar town, Amhara regional state, Ethiopia. The plant material was identified and authenticated by Mr. Getenet Chekole (assistant professor of botanical science) in the Department of Biology, College of Natural and Computational Science, University of Gondar, where specimens with voucher number 01/AT/2022 were deposited in the University of Gondar herbarium. The collected leaves of plant material were washed and dried under shade. The dried material was ground into coarse powder and stored in airtight containers.

## 2.3 Experimental animals

Healthy Swiss albino mice (either-sex) with 25–30 gm and 6–8 weeks of age were obtained from the animal house facilities of the Department of Pharmacology, University of Gondar. These animals were used for the main experiment and the acute dermal toxicity test. The animals were kept in polypropylene cages under standard conditions. One week before the experiment, the animals were acclimatized. All animal handling and care throughout the experiment was as per the National Institute of Health Guidelines for the Care and Use of Laboratory Animals [24].

## 3. Methods

### 3.1 Extraction and fractionation

One kilogram of powdered *C. myricoides* leaves was macerated in an 80% (V/V) methanol solution at room temperature in a round-bottom flask with intermittent shaking for 72 hours. After 72 hours of maceration, it was first filtered through gauze and then through Whatman's No. 1 filters paper. The marc was re-macerated twice in the same manner. The combined filtrate was evaporated using rotary evaporator at 40˚c.

Finally, the concentrated aqueous solution was frozen overnight in a deep freezer and dried using a freeze dryer. The dried powder was weighed, packed in an air tight container, and stored in the refrigerator.

For fractionation, 80 gm extract was suspended in distilled water in a 1:6 ratio using a separatory funnel. To the aqueous suspension, the same amount of n-hexane was added, mixed well, and allowed to form separate distinct layers. The n-hexane fraction was collected in a separate flask, and the fractionation procedure was carried out three times with the addition of the same volume of n-hexane. To obtain ethyl acetate fractions, an equal volume of ethyl acetate to that of distilled water was added to the aqueous fraction. After a distinct layer between the ethyl acetate and aqueous fractions had formed, the ethyl acetate fraction was separated by collecting the bottom aqueous layer and this procedure was repeated twice. The aqueous fraction was concentrated in an oven at 40˚c before being lyophilized, while fractions of all n-hexane and ethyl acetate were evaporated using a rotary evaporator. The fractions were kept in a refrigerator.

### 3.2 Ointment formation

Simple ointment was prepared (Table 1) using a formula described in the British Pharmacopoeia [25]. First, all the ingredients were appropriately measured. These ointment ingredients were poured into the evaporating dish and melted over a water bath according to their descending order of melting points, which are hard paraffin, cetostearyl alcohol, wool fat, and white soft paraffin. The mixture was continuously stirred to achieve homogeneity. Five grams and ten grams of the extract were added to 95 g and 90 g of simple ointment base, respectively, to make 5% and 10% extract ointments. Similarly, 2.5 gm and 5 gm of n-hexane, ethyl acetate, and aqueous fractions were blended with 47.5 g and 45 g of simple ointment base, respectively, to make 50 g of 5% and 10% fraction ointments. For the negative control, 100 g of simple ointment base was taken and treated in the same manner to formulate ointment without an active ingredient.

### 3.3 Acute dermal toxicity

Dermal irritation evaluation was carried out according to OECD Guideline 404 [25]. Since female mice are more sensitive to dermal toxicity than male mice [26], five healthy female Swiss albino mice with normal skin texture aged between 6 and 8 weeks were used. The mice were kept individually in a cage and acclimatized. Before the actual experiment, 10% of the body surface area fur was shaved from the dorsal area of the trunk. After 24 hours, single mouse received 2000 mg/kg of a 10% ointment formulation of the extract. Then it was covered with gauze and a non-occlusive bandage for 24 hours. After 24 hours, all of the materials were carefully removed, and the skin was cleaned with distilled water. The animal was monitored for the development of any unfavorable skin reactions for 24 hours. The test was performed on four additional mice following the same procedure as the single mouse. The animals were observed for 24 hours for any skin reaction. All the animals were examined for the development of any unfavorable skin reactions, in terms of edema and erythema, with close follow-up for 24 hours after washing and then daily for 14 days [27]. The skin reactions, in terms of edema and erythema, were investigated based on the skin reactions scoring system [28, 29].

### 3.4 Grouping and dosing of experimental animals

For the excision and burn wound models, mice were grouped into four; six mice in each group. Group I was vehicle control group, group II and III were treatment groups and group IV was positive control group (nitrofurazone ointment 0.2% w/v). For the incision wound model, five groups of six mice in each group were used. Group I was vehicle control group, group II and III were test groups, group IV positive control group (nitrofurazone ointment 0.2% w/v), and group V was Left untreated (untreated negative control). Efficacy of solvent fractions was examined using eight groups of six mice in each group. Group I was vehicle control group, groups II and III were aqueous fraction treatment group, groups IV and V were ethyl acetate fraction treatment group, groups VI and VII were n-hexane fraction treatment

Table 1. Formula used for the preparation of simple ointment [25].

| Ingredients | Master formula | Reduced formula |
|---|---|---|
| Wool fat | 50 gm. | 5 gm. |
| Hard paraffin | 50 gm. | 5 gm. |
| Cetostearyl alcohol | 50 gm. | 5 gm. |
| White soft paraffin | 850 gm. | 85 gm. |
| Total | 1000 gm. | 100 gm. |

group, group VIII was positive control group (nitrofurazone ointment 0.2% w/v). To evaluate anti-inflammatory activity, animals were assigned into five groups of six mice in each group. Groups I and II were vehicle (2% Tween 80) control and positive control group (indomethacin 10mg/kg), respectively whereas groups III, IV, and V were 100 mg/kg, 200 mg/kg, and 400 mg/kg test extracts, respectively.

## 3.5 Wound healing activity test

**3.5.1 Excision wound model.** Prior to wounds creation, mice were anesthetized with ketamine (80 mg/kg) and diazepam (5 mg/kg) by administering intraperitoneally. The skin fur of the dorsolateral flank area 1–1.5 cm away from the vertebral column on either side and 3 cm away from the ear were shaved and disinfected with 70% alcohol. The anticipated circular wound areas (300 mm$^2$) were marked with a thin permanent marker and created a 2 mm-deep excised wound. The entire wound was left open to the external environment for 2 hours. After recovery, mice were returned to their cage and considered to be on day 0. The simple ointment (vehicle control), extract or solvent fractions, and standard drug (positive control) were applied topically once daily as described in the animal grouping and dosing section, beginning from day one until the wound completely healed in the test group. The wound healing capacities of the crude as well as a solvent fractions were evaluated by the percentage of wound contraction, period of epithelialization, and histological studies [30, 31].

*3.5.1.1 Measurement of wound contraction.* The wound healing progression was evaluated by measuring the rate of wound contraction, which is the percentage reduction of wound size from the origin. It was measured using Piece of transparent paper, often in the form of a grid, is carefully placed over the wound site. The grid helps in measuring the size of the wound until the day of the scab falling off the positive control. The wound healing activities of the extract and solvent fraction were calculated using the initial size of the wound (300 mm$^2$) as 100%, as follows [32].

$$\% \text{ of wound contraction} = \frac{\text{Initial wound size} - \text{wound area on day n}}{\text{Initial wound size}} * 100$$

Where n is the number of days where measurements will be taken (2$^{nd}$, 4$^{th}$, 6$^{th}$, etc.) until complete healing

*3.5.1.2 Epithelialization period measurement.* The epithelialization period was determined as the number of days required for falling off the dead tissue remnants without any residual raw wound [30].

*3.5.1.3 Histopathological analysis.* The skin specimens from each group of excision wound was collected at the end of the experiment after the mice had been sacrificed by an overdose of anesthesia (Ketamine). Skin specimens were kept in 10% buffered formalin, processed, sectioned into 5 $\mu$m, and stained with hematoxylin and eosin. A wound healing process alteration was analyzed by a senior pathologist.

**3.5.2 Incision wound model.** Mice were anesthetized in the same manner as described for the excision wound model. The dorsal fur of each mouse was shaved and decontaminated with 70% alcohol. Longitudinal paravertebral incision (3 cm long and 2 mm deep) was made with a sterile blade on either side at a distance of 1.5 cm from the dorsal midline. Wounds were closed with interrupted sutures, 1 cm apart, with surgical sutures braided silk (no. 00) and left undressed. The wounding day was considered day 0.

The respective treatments (simple ointment, crude extract, and nitrofurazone (0.2%)) were applied once daily topically to the animals of respective groups starting from 24 hours after wound creation (on the 1st day) until the 9th day, and one group was left untreated and served

as an untreated negative control. The stitch was extricated on the eighth day, and the skin-breaking strength of the wound was measured on the tenth day of post-wounding using a continuous water flow technique. Apparatus is set up to measure the tensile strength of the healed wound. The healed wound area attached in two clap forceps in two sides, one side is attached with beaker. The water flow is adjusted to exert a gradual force on the wound, simulating tension or stress. As the water flows over the wound in the beaker, the force required to disrupt or break the healed tissue is measured. This force is indicative of the tensile strength of the wound the higher the force required, the stronger the healed tissue. Breaking strength was compared among the groups. The percentage of tensile strength was calculated as shown below [33]:

$$\text{Percent tensile strength (TS) of the extract or fraction (\%)} = \frac{\text{TS extract} - \text{TS S0}}{\text{TS S0}} * 100$$

$$\text{Percent strength of the positive control (\%)} = \frac{\text{TS positive control} - \text{TS S0}}{\text{TS S0}} * 100$$

$$\text{Percent strength of simple ointment (\%)} = \frac{\text{TS SO} - \text{TS LU}}{\text{TS S0}} * 100$$

Where SO = simple ointment and Lu = left untreated

**3.5.3 Burn wound model.** The mice were anesthetized in the same manner as described for the excision wound model. The dorsal fur of each mouse was shaved and decontaminated with 70% alcohol. The burn wound was created through hot molten beeswax at 80˚c, which was poured into a cylinder with a 300 mm$^2$ circular opening placed on the shaven back of the mice until the wax solidified. The cylinder was removed after approximately 10 to 12 minutes of solidification, leaving the marked circular burn. The animal was placed in a separate cage. Ointment was applied over the wound area with the respective groups as described in the grouping and dosing section every day, starting from day one until the day of the scab falling off the positive control. The progress of healing was examined every 2 days by measuring the percentage of wound contraction and epithelialization [34]. Wound contraction, epithelialization, and histopathological analysis were examined in the same the same manner as described for the excision wound model.

## 3.6 Anti-inflammatory activity

As a model for acute inflammation, carrageenan-induced paw edema was employed. We examined the anti-inflammatory activity of 80% hydrometanolic crude extract of the leaves of *C. myricoides* using Swiss albino mice of either sex. Before administering any medication, each mouse's right hind paw's basal volume was measured using a water displacement plethysmometer (model: PLM 01 PLUS) after an overnight fast with free access to water. Following the determination of the basal volume, the animals were grouped into five, each of which had six mice, with no significant mean volume difference. The mice were given oral doses of 100 mg/kg, 200 mg/kg, and 400 mg/kg of the plant extracts before one hour of induced inflammation. The doses were selected based on the acute oral toxicity test from a previous study [22, 35], along with 10 mg/kg of indomethacin (positive control) and the vehicle (negative control, 2% Tween 80). To induce inflammation, 0.05 ml of a 1% carrageenan in 0.9% saline (w/v) solution was injected into the right hind paw's (sub-plantar) area of mice. Using a plethysmometer, the paw volume was measured at 0, 1, 2, 3, and 4 hours following the injection of the inflammatory stimuli (carrageenan). The percentage of edema inhibition in treated animals was calculated in

comparison to the negative control group. The results obtained were compared among groups [36].

$$\% \text{ Edema inhibition} = \frac{\text{PEC} - \text{PET}}{\text{PEC}} * 100$$

PEC = paw edema control group
PET = paw edema test group

## 3.7 Statistical analysis

All values were presented as mean ± SEM, and statistical analyses were carried out using SPSS software (version 24.0). The result was statistically analyzed using a one-way ANOVA followed by a post-hoc Tukey test. The value of probability less than 5% ($P < 0.05$) was considered statistically significant.

# 4. Results

From the total of 1000 g of course leaf powder of *C. myricoides*, 220 g (22%), 42 g (52.5%), 20 g (25%), and 16 g (20%) extract, distilled water, n-hexane, and ethyl acetate fractions was produced, respectively.

## 4.1 Acute dermal toxicity test

Throughout the 14 day observation, topical treatment of a limited dosage of 2000mg/kg of a 10% extract ointment formulation did not show any signs of inflammation, edema, or erythema.

## 4.2 Evaluation of wound healing

**4.2.1 Excision wound model.** The wound healing progress of mice treated with 5%, 10% ointments and controls is depicted in (Fig 1). Mice treated with 10% and 5% extract ointments showed significant wound contraction ($p < 0.001$) activity starting on the fourth and sixth day as compared negative control, respectively. Treatment with both 10% extract oinment and 0.2% nitrofurazone ointments showed significant wound contraction ($p < 0.001$) starting from the fourth day of treatment as compared to 5% extract ointment. The percentage of wound contractions was increasing over time in all groups. On day 16, negative control (simple ointment), 5% extract ointment, 10% extract ointment, and 0.2% nitrofurazone ointment showed 91.22%, 97.61%, 100%, and 99.66% of wound contractions, respectively (Table 2).

Aqueous fraction at 10% ointment showed a significant wound contraction effect on days 4 ($p < 0.01$) and 6 ($p < 0.001$) compared to vehicle control and 5% EAF, respectively. Treatment with 10% aqueous fraction ointment produced significant wound contraction effect on days 6 ($p < 0.05$), 8 ($p < 0.01$), 10 ($p < 0.001$), and 12 ($p < 0.001$) as compared to 5% aqueous fraction ointment. In addition, the effect of this preparation was comparable to the effects of 0.2% nitrofurazone treatment. Treatment with 5% aqueous fraction ointment showed significant wound contraction on day 6 ($p < 0.05$) and after the 8th day ($p < 0.001$) of treatment as compared to vehicle control. Treatment with 5% and 10% n-hexane fractions showed significant wound contraction on day 8 ($p < 0.001$) as compared to vehicle control (Table 3).

*4.2.1.1 Period of epithelialization.* In comparison to simple ointment, extract ointments at 5% and 10% concentration significantly ($p < 0.001$) reduced the duration of epithelialization period which is comparable to the effect of positive control. Treatments with 10% extract ointment and 0.2% nitrofurazone ointment significantly ($p < 0.05$) shorten the period of

|  | **Simple ointment** | **5% (w/w) extract ointment** | **10% (w/w) extract ointment** | **0.2% (w/v) nitrofurazone ointment** |

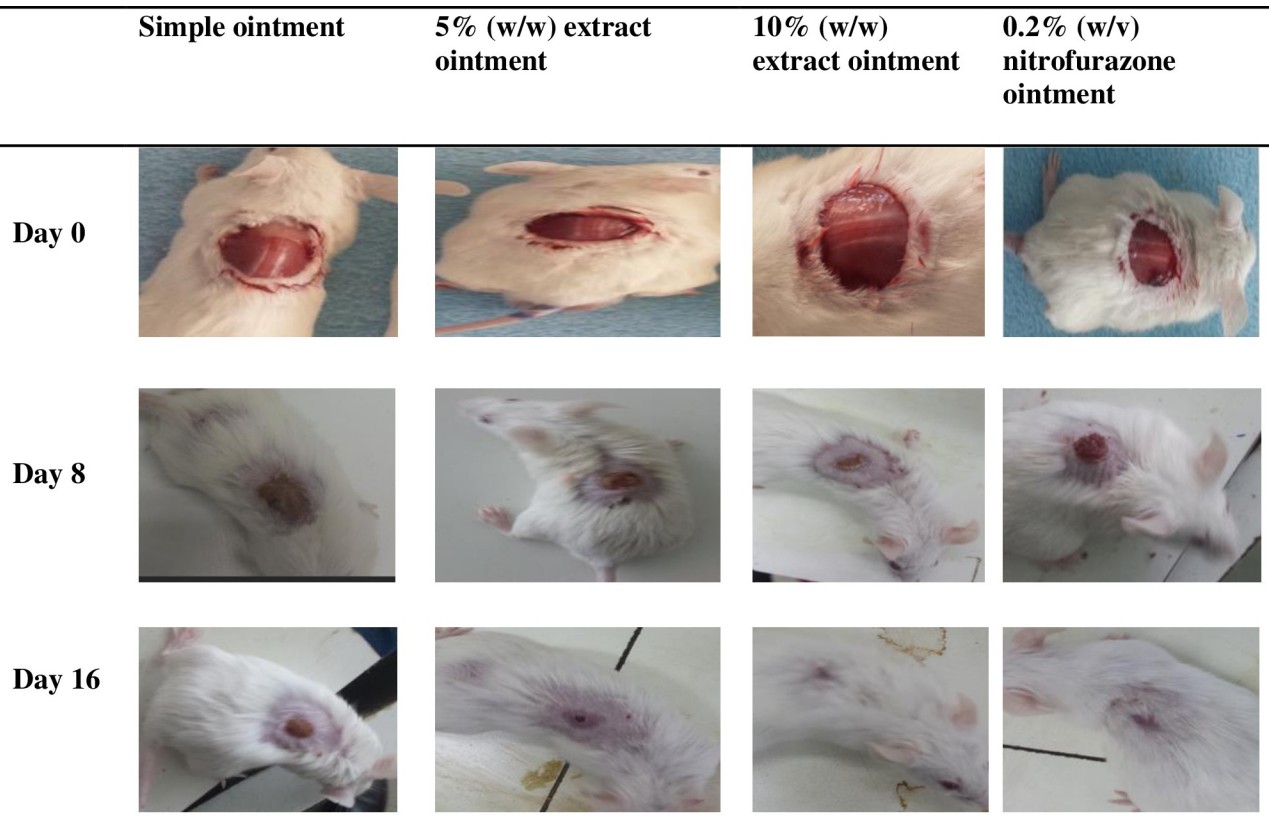

**Fig 1. Photograph of excision wound healing progress after treated with the hydrometanolic extract of the *C. myricoides*.**

epithelialization as compared to the 5% crude extract. The mean period of epithelialization in mice treated with 5% and 10% extract ointments, as well as 0.2% nitrofurazone ointment has been reduced by 13.37%, 27.55%, and 24.38%, respectively, as compared to negative control (Fig 2).

The highest concentration of aqueous fraction showed a significantly short period of epithelialization ($p < 0.001$) as compared to both the negative control and 5% ethyl acetate ointment.

**Table 2. Wound healing activity of extract of the leaves of *Clerodendrum myricoides* on excision wound model in mice.**

| Post-wounding days | Wound area (mm2) ± SEM (% wound contraction) | | | |
|---|---|---|---|---|
|  | **Simple ointment** | **5% (w/w) Extract ointment** | **10% (w/w) Extract ointment** | **0.2% (w/v) Nitrofurazone ointment** |
| Day2 | 282.0±2.449(6.00) | 273.16±3.26(8.94) | 276.33±1.94(7.88) | 273.83±1.86(8.72) |
| Day4 | 253.83±3.919(15.38) | 245.83±3.58(18.04) | 216.5±2.99(27.83)(ab)[3] | 220.50±2.98(26.5)(ab)[3] |
| Day6 | 214.16±3.97(28.611) | 167.5±5.93(44.16)a[3] | 138.00±1.57(54.00) (ab)[3] | 142.83±2.05(52.3) a[3]b[2] |
| Day8 | 185.16±3.73(38.27) | 103.16±6.48(65.60)a[3] | 82.33±2.66(72.55)a[3]b[1] | 81.16±6.5(72.94) a[3]b[1] |
| Day10 | 118.83±3.17(60.38) | 63.33±4.46(78.88)a[3] | 48.5±1.47(83.83) a[3]b[1] | 47.00±1.91(84.33) a[3]b[2] |
| Day12 | 87.5±2.75(70.83) | 35.5±3.67(88.16)a[3] | 18.83±1.49(93.72) a[3]b[2] | 22.00±1.18(92.66) a[3]b[2] |
| Day14 | 53.16±2.97(82.27) | 16.66±1.22(94.44)a[3] | 5.16±1.19(98.27) a[3]b[2] | 5.66±1.60(98.11) a[3]b[2] |
| Day16 | 26.33±1.92(91.22) | 7.16±0.74(97.61)a[3] | 0.00±0.00(100.00) a[3]b[2] | 1.00±0.63(99.66) a[3]b[2] |

Values are expressed as mean ± SEM ($n$ = 6), and one-way ANOVA followed by post hoc Tukey test were used for analysis. [a]Compared to simple ointment [b]compared to 5% (w/w) extract ointment. [1]$p < 0.05$, [2]$p < 0.01$, [3]$p < 0.001$. Initial wound area was 300 mm$^2$.

**Table 3. Wound healing activity of solvent fractions of the leaves of *Clerodendrum myricoides* on excision wound model in mice.**

| | Wound area (mm2) ± SEM (% contraction) | | | | | | | |
|---|---|---|---|---|---|---|---|---|
| | Simple ointment | 5% (w/w) EAF ointment | 10% (w/w) EAF ointment | 5% (w/w) NHF ointment | 10% (w/w) NHF ointment | 5% (w/w) AQF ointment | 10% (w/w) AQF ointment | 0.2% (w/v) NF ointment |
| Day2 | 273.83±2.04 (8.72) | 272.83±2.63 (9.06) | 270.67±2.11 (9.78) | 274.50±3.92 (8.5) | 271.67±3.26 (9.44) | 269.83±3.26 (10.06) | 265.33±4.29 (11.56) | 264.00±4.52 (12.00) |
| Day4 | 255.83±1.47 (14.72) | 248.83±1.85 (17.06) | 248.17±1.45 (17.28) | 247.33±2.51 (17.56) | 245.83±3.07 (18.06) | 246.33±2.72 (17.89) | 240.33±2.47 (19.89)a$^2$ | 238.83±2.36 (20.39)a$^3$ |
| Day6 | 211.50±1.77 (29.5) | 205.67±2.04 (31.44) | 201.83±1.58 (32.72) | 202.67±2.49 (32.44) | 201.67±3.03 (32.56) | 198.50±2.93 (33.83)a$^1$ | 187.17±2.75 (37.61)(ab)$^3$ (cde)$^2$ f$^1$ | 184.48±2.47 (38.50)(abcde)$^3$ f$^2$ |
| Day8 | 171.67±1.91 (42.78) | 155.83±2.23 (48.06)a$^2$ | 152.17±1.54 (49.28) a$^2$ | 152.50±2.43 (49.17) a$^3$ | 151.00±2.86 (49.50) a$^3$ | 146.83±2.80 (51.06)a$^3$ | 132.18±2.91 (55.94) (abcde)$^3$ f$^2$ | 129.50±2.47 (56.83)(abcdef)$^3$ |
| Day10 | 131.67±1.91 (56.11) | 113.66±1.90 (62.11)a$^3$ | 106.33±1.14 (64.56) a$^3$ | 105.33±3.19 (64.89) a$^3$ | 101.20±3.01 (66.17) a$^3$b$^1$ | 96.67±3.05 (67.78)a$^3$b$^2$ | 77.00±2.76 (74.33) (abcdef)$^3$ | 74.50±2.50 (75.22) (abcdef)$^3$ |
| Day12 | 82.16±1.66 (72.61) | 63.66±2.18 (78.78) a$^3$ | 59.67±1.44 (80.11) a$^3$ | 54.16±3.23 (81.94) a$^3$ | 51.16±2.90 (82.89) a$^3$b$^1$ | 49.33±3.31 (83.56) a$^3$b$^2$ | 31.33±2.75 (89.56)(abcdef)$^3$ | 29.49±2.49 (90.22) (abcdef)$^3$ |
| Day14 | 55.00±1.55 (81.67) | 32.16±1.93 (89.28) a$^3$ | 27.50±1.77 (90.78) a$^3$ | 23.16±3.22 (92.28) a$^3$ | 20.33±3.10 (93.22) a$^3$b$^1$ | 16.33±3.02 (94.56) a$^3$b$^2$ | 6.57±2.66 (97.83)(abc)3d$^2$e$^1$ | 4.50±2.43 (98.50)(abcd)$^3$e$^2$f$^1$ |
| Day16 | 28.00±1.55 (90.67) | 14.33±1.38 (95.22) a$^3$ | 10.33±1.82 (96.56) a$^3$ | 8.83±1.90 (97.06) a$^3$ | 7.00±2.14 (97.67) a$^3$b$^1$ | 3.50±1.69 (98.83) a$^3$b$^3$ | 0.00±0.00 (100)(ab)$^3$ (cd)$^2$e$^1$ | 0.00±0.00 (100.00)(ab)$^3$cd$^2$e$^1$ |

Values are expressed as mean ± SEM ($n = 6$) and one-way ANOVA followed by post hoc Tukey's test were used for analysis; [a] compared to simple ointment; [b] compared to 5% w/w EAF, [c] compared to 10% w/w EAF; [d] compared to 5% w/w NHF; [e] compared to 10% w/w NHF; [f] compared to 5% AQF. [1] $p<0.05$, [2] $p<0.01$, and [3] $p<0.001$. Initial wound area was 300 mm$^2$, where AQF: aqueous fraction; EAF: ethyl acetate fraction: NHF: n-hexane fraction; NF: Nitrofurazone

Both 5% and 10% n-hexane fraction ointments showed significantly short periods of epithelialization ($p<0.001$) as compared to negative control. Moreover, 5% and 10% ethyl acetate ointments significantly reduced the period of epithelialization ($p<0.01$) as compared to negative control. The highest percentage of epithelialization period (26.82%) reduction is produced by 0.2% nitrofurazone ointment while the lowest percentage of epithelialization period (10.58%) reduction is produced by 5% ethyl acetate fraction (Table 4).

*4.2.1.2 Histopathological analysis.* In the excision wound model, wounds of mice treated with 10% extract ointment and 0.2% nitrofurazone ointment showed, high fibroblast proliferation, and collagen deposition compared to the negative control (simple ointment). However, wounds treated with 5% extract ointment showed modest fibroblast proliferation and collagen deposition. On the other hand, moderate and mild wound neovascularization were also observed in 5% and 10% extract ointment treatment groups, respectively (Table 5 and Fig 3).

**4.2.2 Incision wound model.** The tensile strength (wound-breaking strength) of the incision wound was measured on day ten. The tensile strength was significantly ($p<0.001$) higher in the 10% crude extract and 0.2 nitrofurazone ointment treated groups as compared to the untreated, vehicle, and 5% extract ointment treated groups. Whereas, tensile strength was significantly higher ($p<0.01$) with 5% extract ointment treated groups as compared to untreated and vehicle treated groups. Relative to the vehicle treated (simple ointment) group, the percent increase of the tensile strengths were 22.39, 57.56, and 63.66% in the 5%, 10%, and 0.2% nitrofurazone ointment treated groups, respectively (Table 6 and Fig 4).

**4.2.3 Burn wound model.** Extract at 10% ointment showed significant wound contraction ($p<0.001$) starting from the eighth day of treatment as compared to negative control (simple ointment) while, 5% extract ointment reached significant wound contraction level on day 10 ($p<0.01$). On the other hand, treatment with positive control (0.2% nitrofurazone) showed significantly higher wound contraction effect starting from day 6 ($p<0.01$) to the last day of

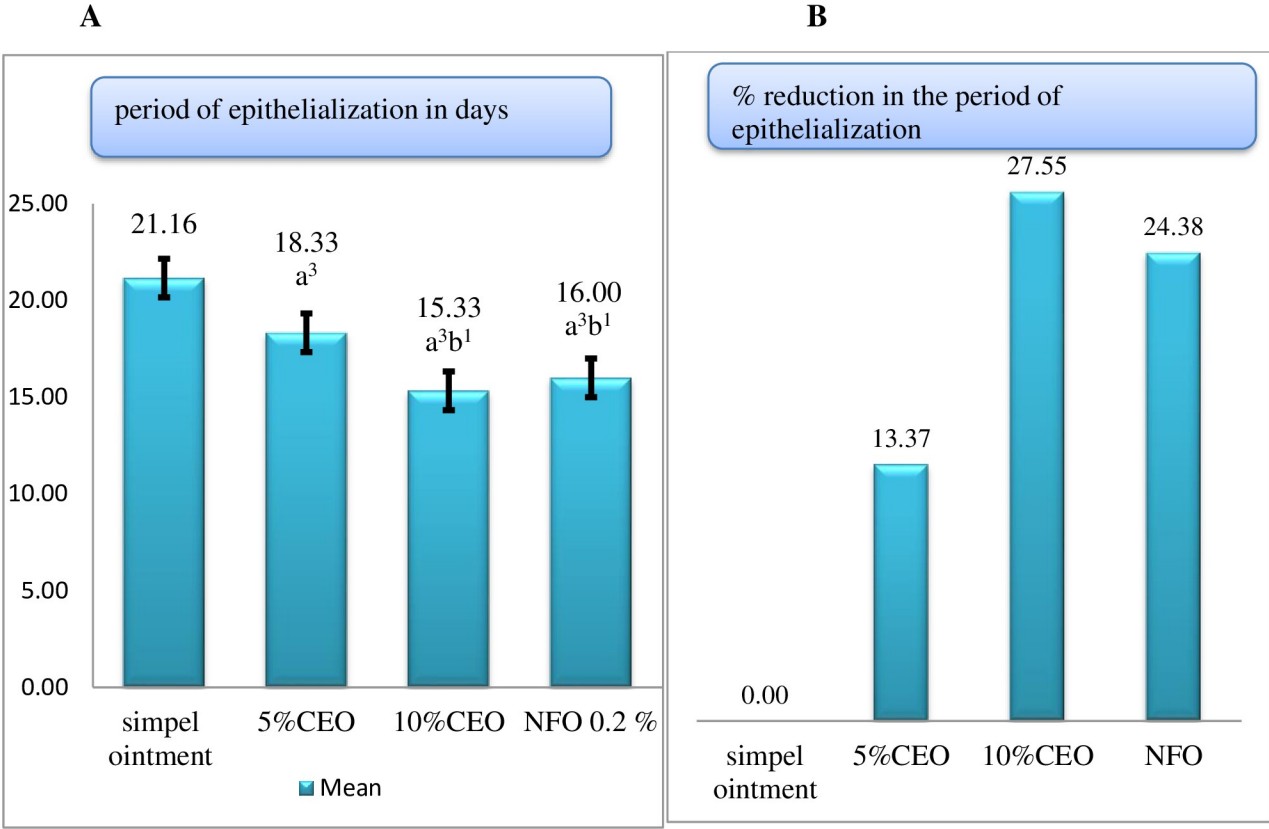

**Fig 2. Effect of extract on period of epithelialization in excision wound.** (A) = period of epithelialization, (B) = Percentage reduction in the period of epithelialization. Values are expressed as mean ± SEM (n = 6)and one-way ANOVA followed by post hoc Tukey's test were used for analysis. [a] Compared to simple ointment, [b] compared to 5% CEO, [1]$p < 0.05$, [3]$p < 0.001$, percent reduction in period of epithelialization is from the SO treated group. Where CEO: crude extract ointment; NFO: nitrofurazone ointment.

observation (20[th] day) ($p < 0.001$) as compared to both negative control (simple ointment) and 5% extract ointment (Table 7 and Fig 5).

*4.2.3.1 Period of epithelialization.* The epithelialization period was significantly shorter in groups treated with 5% extract ointment ($p < 0.05$), 10% extract ointment ($p < 0.001$), and 0.2%

**Table 4. Wound healing activity of extract of the leaves of *Clerodendrum myricoides* solvent fraction on period of epithelialization.**

| Treatment group | Mean period of epithelialization in days | % decrease in the period of epithelialization |
|---|---|---|
| Simple ointment | 20.50±0.76 | - |
| 5% (w/w) EAF ointment | 18.33±0.33a[2] | 10.58 |
| 10% (w/w) EAF ointment | 18.00±0.26a[2] | 12.19 |
| 5% (w/w) NHF ointment | 17.33±0.21a[3] | 15.46 |
| 10% (w/w) NHF ointment | 17.33±0.21a[3] | 15.46 |
| 5% (w/w) AQF ointment | 16.67±0.21a[3] | 18.68 |
| 10% (w/w) AQF ointment | 15.33±0.42(ab)[3] c[2] (de)[2] | 25.21 |
| 0.2% (w/v) Nitrofurazone ointment | 15.00±0.45(abc)[3] (de)[2] | 26.82 |

Values are expressed as mean ±SEM (n = 6), and one-way ANOVA followed by a post hoc Tukey test was used for analysis. [a] Compared to simple ointment; [b] compared to 5% (w/w) EAF ointment; [c] compared to 10% (w/w) EAF ointment; [d] compared to 5% (w/w) NHF ointment; [e] compared to 10% (w/w) NHF ointment. [2]$p < 0.01$; [3]$p < 0.001$. Where AQF: aqueous fraction; EAF: ethyl acetate fraction; NHF: n-hexane fraction.

**Table 5. Effect of the leaves of *Clerodendrum myricoides* hydroalcoholic extract on the excision wound histology of skin of mice.**

| Group | PoF | DoC | PCs | MCs | N |
|---|---|---|---|---|---|
| Vehicle control | ↑ | ↑ | - | ↑ | - |
| 5% (w/w) *Clerodendrum myricoides* Extract | ↑ | ↑↑ | - | ↑ | ↑ |
| 10% (w/w) *Clerodendrum myricoides* Extract | ↑↑ | ↑↑↑ | ↑ | ↑↑ | ↑↑ |
| 0.2% (w/v) Standard drug | ↑↑ | ↑↑↑ | ↑ | ↑↑ | ↑↑ |

Notes: Low concentration (↑), moderate concentration (↑↑), high concentration (↑↑↑), absent (-). Abbreviations: PoF, Proliferation of Fibroblast; DoC, Depositions of Collagen; MCs, Mononuclear Cells; PCs, Polymorphnuclear Cells; N, Neovascularization

nitrofurazone ointment (p<0.001), as compared to the negative control. In addition, in 0.2% nitrofurazone and 10% extract ointment treated groups the period of epithelialization was significantly shortened as compared to the 5% extract treatment ointment treated group. The percent of shortening of the periods of epithelialization by 5% extract, 10% extract, and 0.2% nitrofurazone ointments were 8.76%, 16.77%, and 17.52%, respectively (Fig 6).

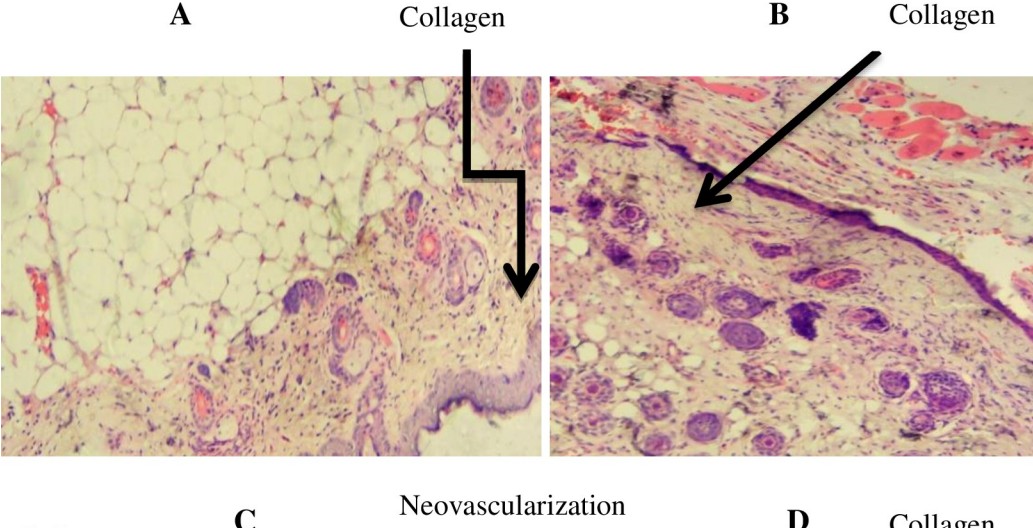
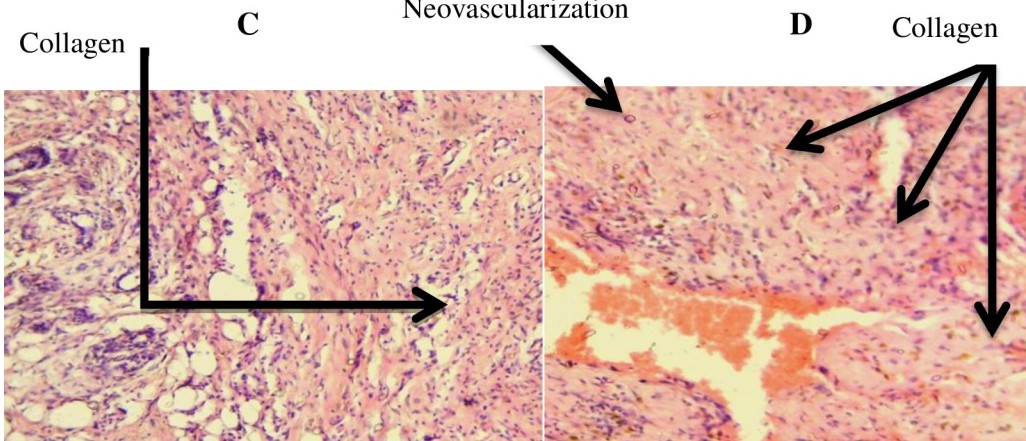

**Fig 3. Images of histological sections of excision wound tissue.** Notes: (A): histological section of wound tissue from simple ointment, (B): histological section of wound tissue of 5% extract treated mouse, (C): histological section of wound from 10% extract treated mouse, (D): histological section of wound from nitrofurazone 0.2% treated mouse.

**Table 6. Wound healing activity of extract of the leaves of *Clerodendrum myricoides* on tensile strength of incision wound model.**

| Group | Mean tensile strength ± SEM | Percent increase of tensile strength |
|---|---|---|
| Untreated | 210.16±(6.94) | - |
| Simple ointment | 218.00±(1.69) | 3.73 |
| 5% (w/w) extract ointment | 266.83±(5.23)a³b² | 22.39 |
| 10% (w/w) extract ointment | 343.5±(12.7)(abc)³ | 57.56 |
| 0.2% (w/v) Nitrofurazone ointment | 356.8± (9.32)(abc)³ | 63.66 |

Values are expressed as mean ± SEM ($n$ = 6), and one-way ANOVA followed by post hoc Tukey test was used for analysis. [a]Compared to the untreated group; [b]compared to the simple ointment-treated group; [c]compared to the 5% extract ointment-treated group. [2]$p<0.01$, [3]$p<0.001$.

*Histopathological analysis.* In burn wound model, moderate fibroblast proliferation and high collagen deposition were observed in mice treated with 10% extract and 0.2% nitrofurazone ointments. On the other hand, moderate fibroblast proliferation and low collagen deposition were observed in mice treated with 5% ointment. High amounts of mononuclear inflammatory cells were also seen in 5% extract ointment treated groups (Table 8 and Fig 7).

## 4.3 Evaluation of anti-inflammatory activity

The extract containing 200 mg/kg leaves of *Clerodendru mmyricoides* showed a significant ($p<0.05$) reduction in paw edema starting from one hour as compared to a negative control. Similarly, the extract containing 400 mg/kg and indomethacin 10mg/kg showed a significant ($p<0.001$) reduction in paw edema activity starting after one hour as compared to the negative control. Extracts containing 200 mg/kg and 400 mg/kg, as well as indomethacin 10 mg/kg, showed a significant ($p<0.001$) inhibition of edema after two hours of carrageenan injection as compared to a negative control and 100 mg/kg extract. On the other hand, the lowest dose (100 mg/kg) of the extract doesn't show any significant reduction in paw edema up to a four-hour follow-up. The maximum percent of inflammation inhibition is shown in 4 hours by 400 mg/kg extract (44.44%) and standard (50.90%) (Table 9 and Fig 8).

| A | B | C |
|---|---|---|

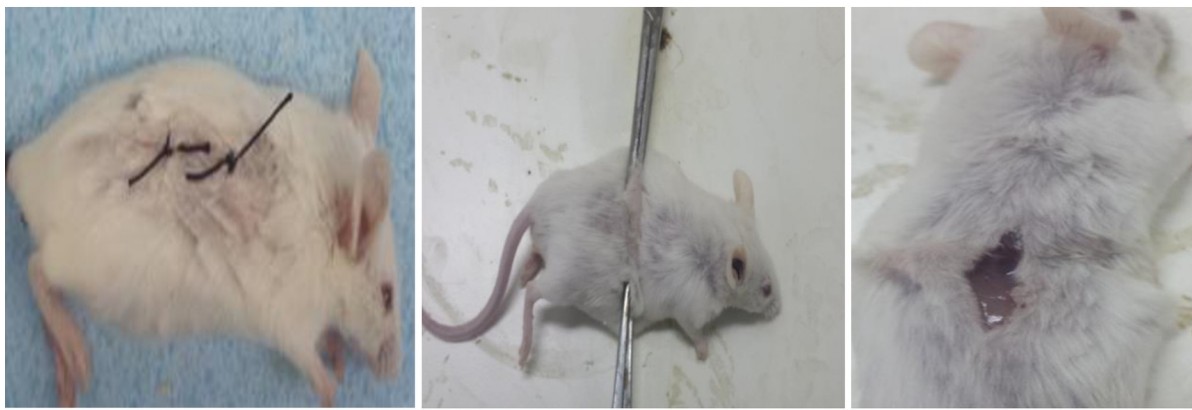

**Fig 4. Photograph of incision wound.** Note: (A): on the day of wound creation, (B): measuring tensile strength by water flow method, (C): broken skin by water flow technique.

**Table 7. Wound healing activity of extract of the leaves of *Clerodendrum myricoides* on burn wound model in mice.**

| Post-wounding days | Wound area (mm2) ± SEM (% wound contraction) | | | |
| --- | --- | --- | --- | --- |
| | Simple ointment | 5% (w/w) extract ointment | 10% (w/w) extract ointment | 0.2% (w/v) nitrofurazone ointment |
| Day2 | 290.00±1.34(3.33) | 288.00±1.15(4.00) | 287.83±1.87(4.06) | 287.00±1.86(4.33) |
| Day4 | 280.50±2.14(6.50) | 274.50 ±1.09(8.50) | 274.33±1.67(8.50) | 274.00±1.98(8.06) |
| Day6 | 267.50±2.03(10.83) | 264.67±1.71(11.78) | 261.00±1.48(13.00) | 255.83±1.68(14.72)(ab)[2] |
| Day8 | 256.67±1.33(14.44) | 250.67±1.45(20.06) | 239.83±1.89(20.06) a[3]b[2] | 235.50±1.95(18.11)(ab)[3] |
| Day10 | 203.50±2.68(32.17) | 190.17±3.30(39.94)a[2] | 148.00±2.56(50.67)(ab)[3] | 140.50±0.92(53.17)(ab)[3] |
| Day12 | 157.83±5.53(47.39) | 136.17±4.70(54.61)a[2] | 99.33±1.82(66.89)(ab)[3] | 94.00±2.84(68.67)(ab)[3] |
| Day14 | 114.33±3.24(61.89) | 99.50±2.03(66.83) a[2] | 70.17±1.96(76.61)(ab)[3] | 63.00±3.34(79.00)(ab)[3] |
| Day16 | 88.00±3.68(70.67) | 67.83±1.51(77.39) a[3] | 21.33±1.56(92.89)(ab)[3] | 15.33±2.81(94.89)(ab)[3] |
| Day 18 | 47.33±3.63(84.22) | 23.83±1.58(92.06) a[3] | 4.50±2.08(98.50) (ab)[3] | 4.83±2.59(98.39) (ab)[3] |
| Day 20 | 23.33±2.09(92.22) | 7.33±2.73(97.56) a[3] | 0.00±0.00(100.00) a[3]b[1] | 0.00±0.00(100.00) a[3]b[1] |

Values are expressed as mean ± SEM ($n$ = 6), and one-way ANOVA followed by post hoc Tukey test were used for analysis. [a]Compared to simple ointment; [b]compared to 5% (w/w) extract ointment. [1]$p<0.05$, [2]$p<0.01$, [3]$p <0.001$. Initial wound area was 300 mm$^2$.

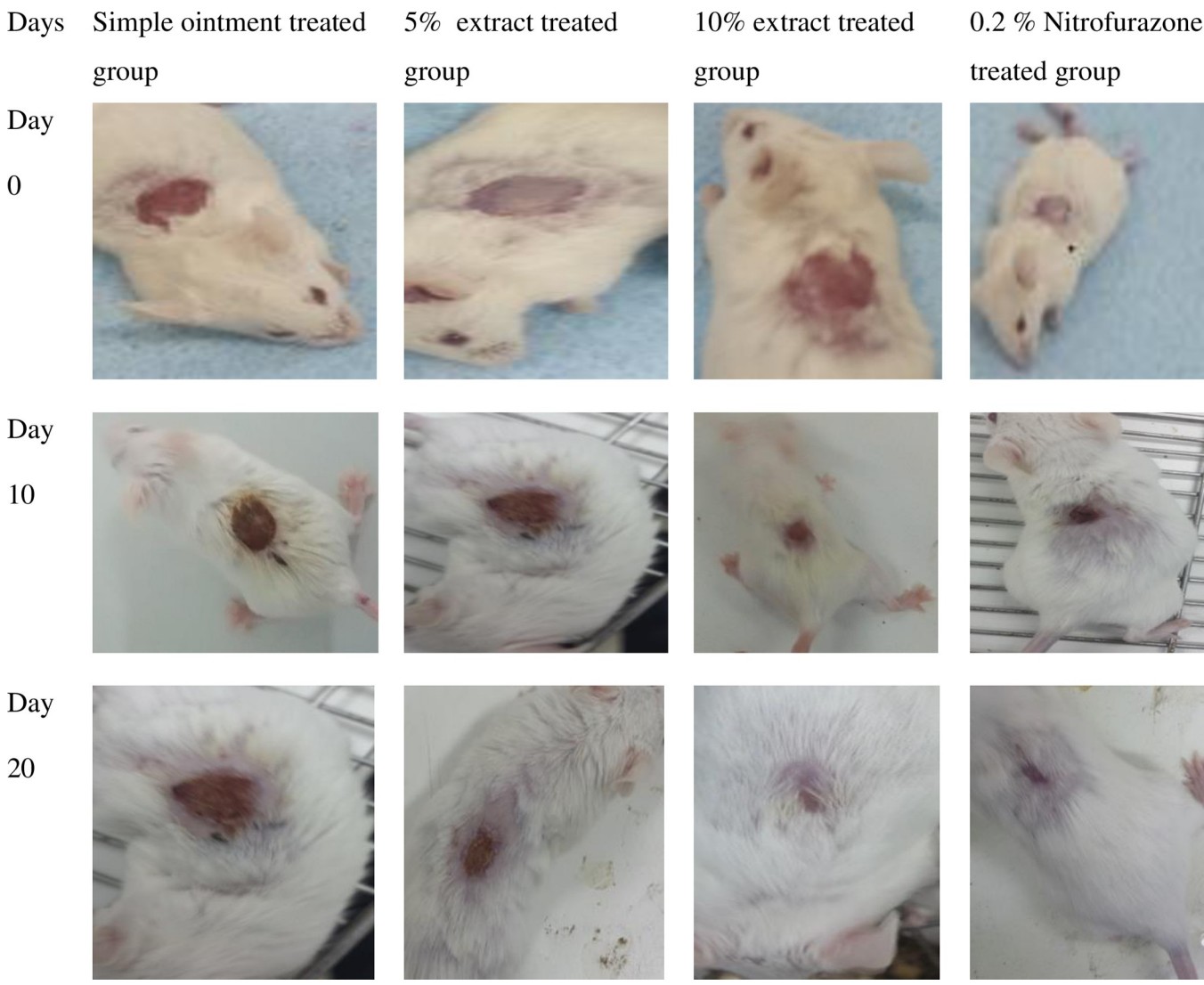

| Days | Simple ointment treated group | 5% extract treated group | 10% extract treated group | 0.2 % Nitrofurazone treated group |

Day 0

Day 10

Day 20

**Fig 5. Photograph of burn wound healing progress after treated with the hydrometanolic extract of the *C. myricoides*.**

A                                                                    B

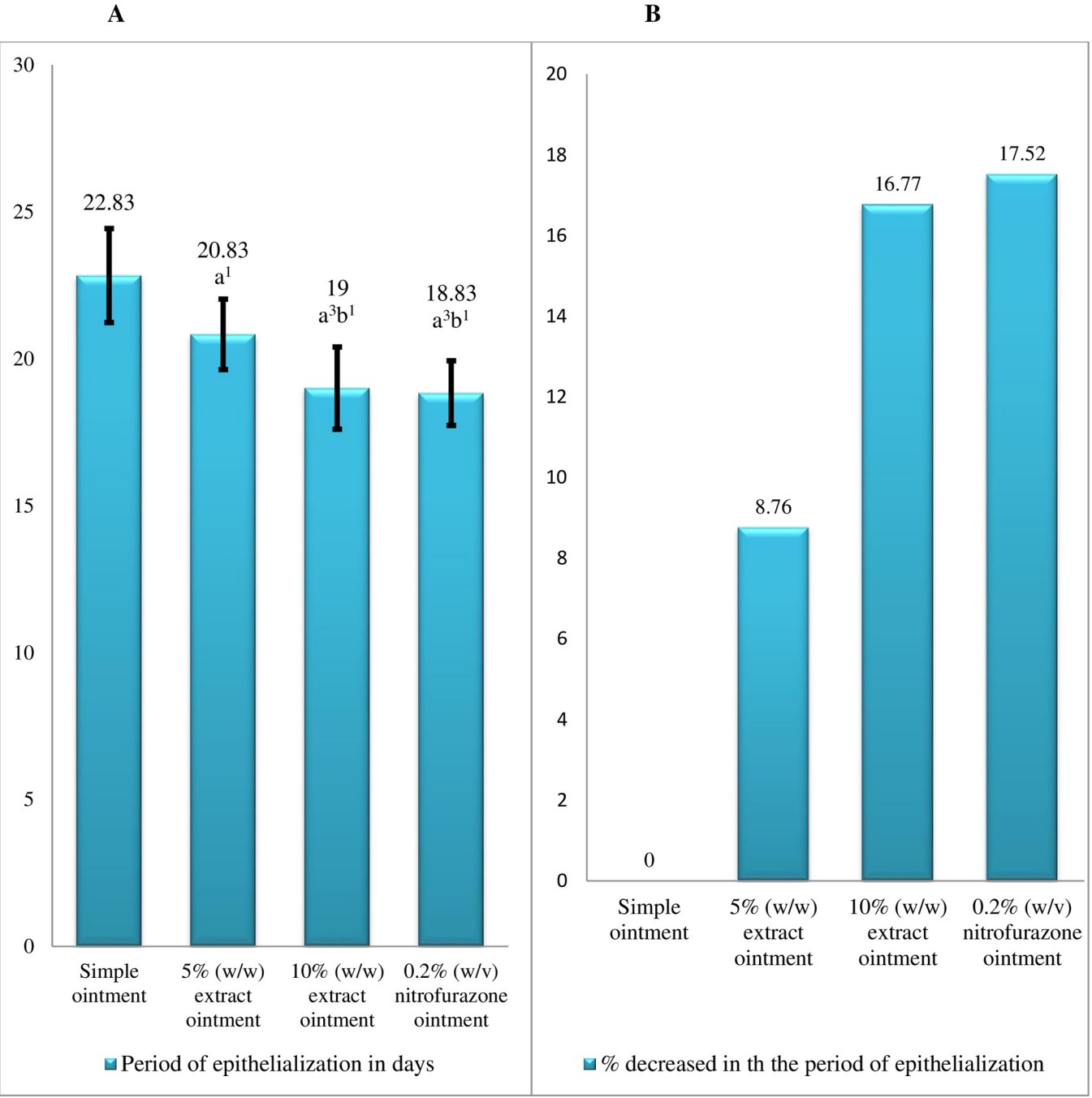

**Fig 6. Effect of extract on period of epithelialization in burn wound.** (A) = period of epithelialization, (B) = Percentage reduction in the period of epithelialization. Values are expressed as mean ± SEM (n = 6)and one-way ANOVA followed by post hoc Tukey's test were used for analysis. aCompared to simple ointment, bcompared to 5% extract, $1p < 0.05$, $3p < 0.001$, percent reduction in period of epithelialization is from the simple ointment treated group.

## 5. Discussion

The findings of this study revealed that 80% methanol extract and solvent fractions of the leaves of *C. myricoides* have wound healing activity. This wound-healing effect of *C. myricoides* leaves extract and solvent fractions might be due to intervening in one or more steps of wound healing. Previous studies carried out on the related species, *C. infortunatum* [37] *and C.*

**Table 8. Effect of the leaves of *Clerodendrum myricoides* hydroalcoholic extract on the burn wound histology of skin of mice.**

| Group | PoF | DoC | PCs | MCs | N |
|---|---|---|---|---|---|
| Simple ointment | ↑ | ↑ | - | ↑ | - |
| 5% (w/w) *Clerodendrum myricoides* extract | ↑↑ | ↑ | ↑ | ↑↑↑ | ↑↑ |
| 10%(w/w) *Clerodendrum myricoides* extract | ↑↑ | ↑↑↑ | ↑ | ↑↑ | ↑↑ |
| 0.2%(w/v) standard drug | ↑↑ | ↑↑↑ | ↑ | ↑↑ | ↑↑↑ |

Low concentration (↑), moderate concentration (↑↑), high concentration (↑↑↑), absent (-), Abbreviations: PoF, Proliferation of Fibroblast; CD, Depositions of Collagen; MCs, Mononuclear Cells; PCs, Polymorphnuclear Cells; N, Neovascularization

*serratum* [38] of the genus *Clerodendrum*, reported significant wound healing activity, suggesting the genus *Clerodendrum* is endowed with wound healing activity.

Since wound healing is an intricate process, it cannot be comprehended by relying just on one model or an in *vitro* experiment. The use of two or more distinct *in vivo* models is necessary for a better understanding of the healing process [39]. Hence, in the present study, three different wound models (excision, incision, and burn) and one anti-inflammatory model (carrageenan-induced mouse paw edema) were used to determine the ability of the leaves extract of *C. myricoides* effect on wound healing *in vivo*.

The wound contraction rate in the excision wound model on day 16 is 100% by 10% crude extract, which is comparable to earlier studies [37]. These might be due to the bioactive

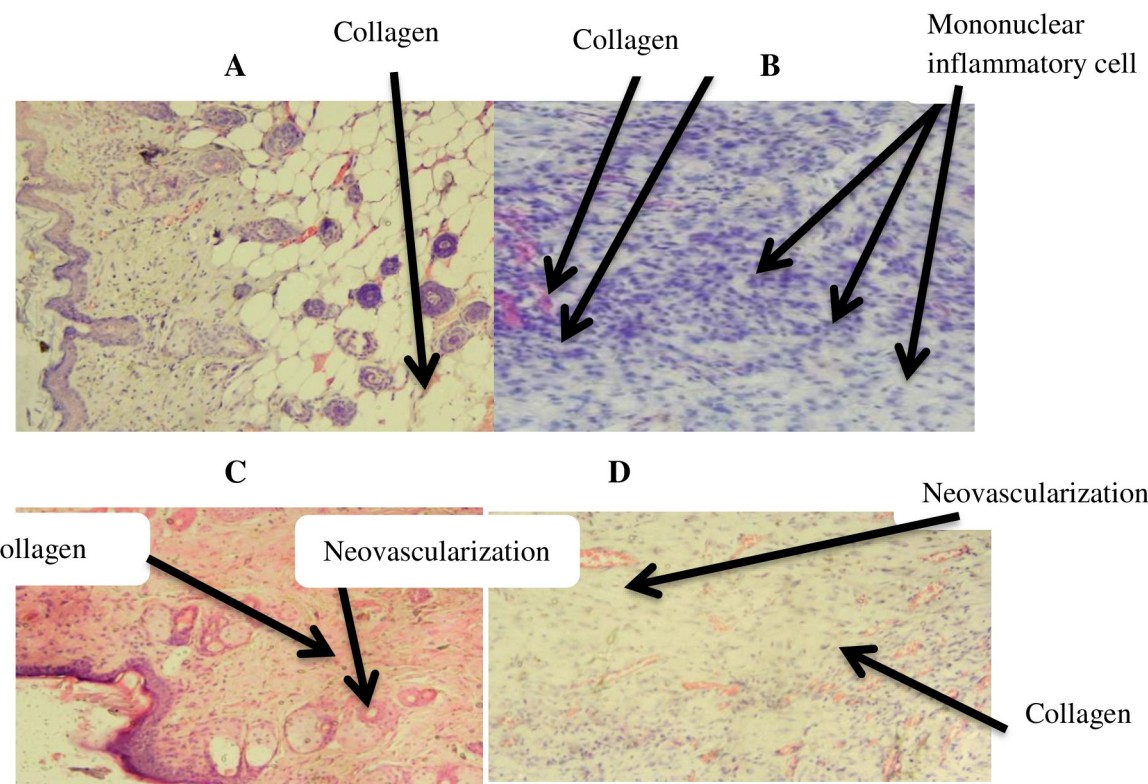

**Fig 7. Images of histological sections of burn wound tissue.** Notes: (A): histological section of wound tissue from simple ointment, (B): histological section of wound tissue of 5% extract treated mouse, (C): histological section of wound from 10% extract treated mouse, (D): histological section of wound from nitrofurazone 0.2% treated mouse.

**Table 9. Anti-inflammatory activity of extract of the leaves of *Clerodendrum myricoides*on carrageenan-induced mice paws edema.**

| | Edema Volume (mL)/ (Percent Edema Inhibition) | | | | |
|---|---|---|---|---|---|
| | 0hr | 1hr | 2hr | 3hr | 4hr |
| -ve control | 0.258±0.010 | 0.332±0.013 | 0.383±0.012 | 0.392±0.00 | 0.387±0.008 |
| 100mg/kg Extract | 0.263±0.010 | 0.328±0.011 (1.2) | 0.368±0.011 (3.91) | 0.365±0.012 (6.88) | 0.365±0.012 (5.68) |
| 200mg/kg Extract | 0.267±0.013 | 0.290±0.006 (12.65)$a^1$ | 0.268±0.03 (30.02) (ab)$^3$ | 0.237±0.00 (39.54) (ab)$^3$ | 0.233±0.006 (39.79) (ab)$^3$ |
| 400mgkg Extract | 0.263±0.009 | 0.265±0.008 (20.18)$a^3b^2$ | 0.250±0.009 (34.72) (ab)$^3$ | 0.230±0.011 (41.32) (ab)$^3$ | 0.215±0.009 (44.44) (ab)$^3$ |
| Indomethacin (10mg/kg) | 0.262±0.10 | 0.255±0.008 (23.19) (ab)$^3$ | 0.242±0.007 (36.81) (ab)$^3$ | 0.213±0.008 (45.66) (ab)$^3$ | 0.190±0.006 (50.90) (ab)$^3c^1$ |

Values are expressed as mean ± SEM (n = 6) and data were analyzed by one way ANOVA followed by post hoc Tukey test; [a]against–ve control; [b] against 100mg/kg, [c] against 200mg/kg; [1]$p < 0.05$, [2]$p < 0.01$, [3]$p < 0.001$.

compounds found in the plant extract. Herbal extracts may speed up the wound healing process by their anti-inflammatory, anti-bacterial, and anti-oxidant activities [40].

Wound healing activity of *C. myricoides* might be related to the ability of extracts to increase fibroblast proliferation, collagen production, and anti-inflammatory action. This is in line with other study findings, which indicate that *C. infortunatum* can speed up wound healing in diabetic rats by speeding cell proliferation, cell migration, and the extract's anti-inflammatory properties [41]. This is also supported by the result that the leaves of *C. myricoides* extract displayed enhanced concentrations of fibroblasts and collagen depositions, as well as a low or

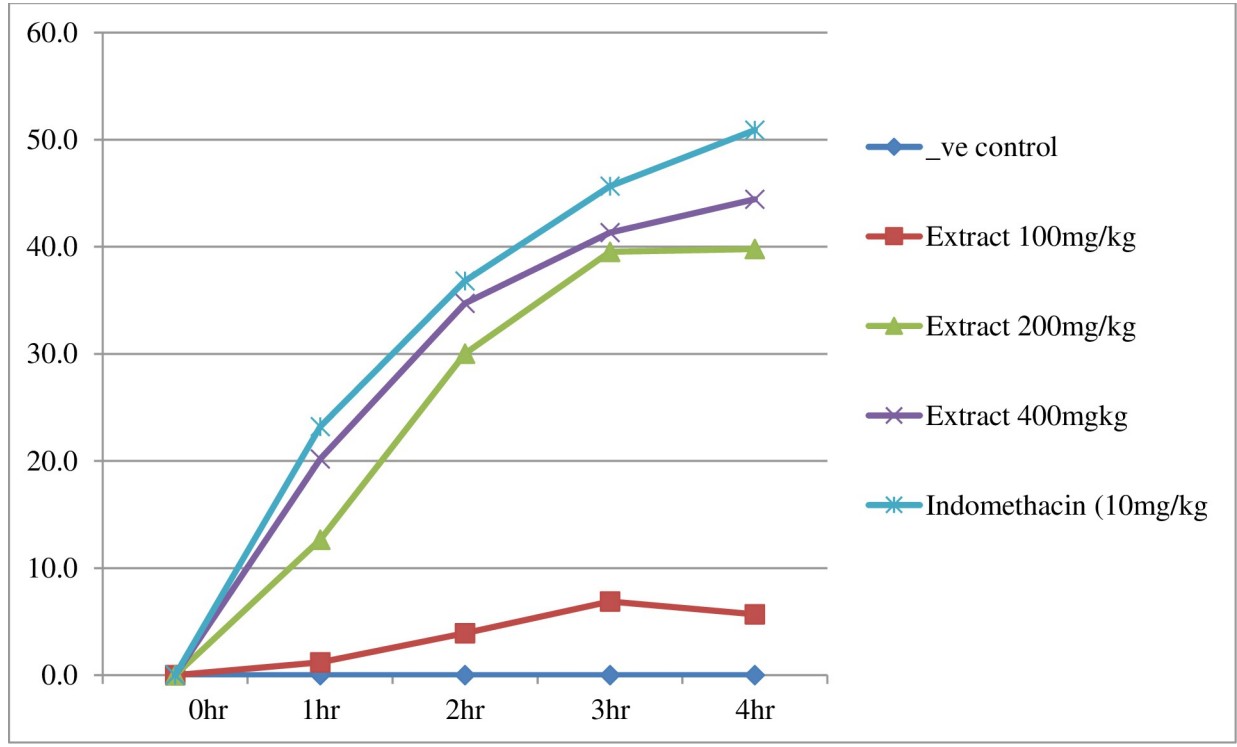

**Fig 8. Percent of inflammation inhibition of extract of the leaves of *Clerodendrum myricoides* on carrageenan-induced mice paws edema.**

absence of inflammatory cell depositions in specimens from treatment groups, according to a histopathologic analysis of the samples.

In this study, the leaves of *C. myricoides* extract showed a significant reduction in the period of epithelialization. Since wound contraction minimizes the size of the wound and the amount of ECM required to fix the defect, it speeds up the healing process. Contraction also makes re-epithelialization easier by reducing the migration distances keratinocytes have to travel [42]. Therefore, the faster epithelialization process of the wound following 80% methanol extract administration might be due to epithelial cell proliferation that is promoted or increased viability [43].

A prior investigation disclosed that a methanol extract of *C. myricoides* leaves exhibited strong antioxidant properties [20, 44]. Oxidants scavenge healthy cells in the vicinity of the wound and attack macromolecules include lipid, protein, and DNA, which causing cellular/tissue damage [45]. Matrix metalloproteinase can also be produced in greater amounts in the presence of excessive oxidants. By destroying extracellular matrix components, especially collagens, excessive MMPs, in particular collagenases, slow the healing of wounds [46]. The antioxidant properties of the *C. myricoides* extract may play a role in wound healing activity. This is in line with a study on the ethanol extract of *C. infortunatum*, which showed strong antioxidant and wound healing activities [47].

Studies also showed that different solvent extracts (petroleum ether, chloroform, acetone, and methanol) of *C. myricoides* leaves showed antimicrobial activities against common wound-infecting microbes, such as *Staphylococcus aureus*, *methicillin-resistant Staphylococcus aureus*, *Shigella sonnei*, and *E. coli* [18, 19]. Bacteria can impede the healing of wounds by prolonged elevation of TNF-α and pro-inflammatory cytokines such as interleukin-1 (IL-1) and elongate the inflammatory phase. This may lead the wound to chronic non healing state and may fail to heal. The antibacterial properties of *C. myricoides* extract shown in the previous study might be another mechanism that could be used to explain the wound healing effect of the extract. A previously conducted investigation on *pupalia lappaceajuss* which described that the plant's antibacterial activity facilitates wound healing [48].

The aqueous fraction of *C. myricoides* showed superior activity. The findings of this study might be explained by the fact that the majority of the plant's bioactive components involved in wound healing activity are water-soluble in nature. To extract more polar chemicals, including flavonoids, glycosides, tannins, and certain alkaloids, polar solvents like water are favorable [49]. Because of their astringent and antibacterial qualities, alkaloids, tannins, and flavonoids are known to speed up the healing of wounds [50]. In different to this, there was no discernible difference between the groups treated with the ethyl acetate and n-hexane fractions in terms of wound contraction rate up to 10 days or time of epithelialization. This might be due to the fact that the majority bioactive component compositions that have wound healing potential may found on aqueous fraction, which could be the reason for this.

The breaking strength in incision wounds provided additional evidence of the crude extract's superior effectiveness in wound healing. Better wound healing is indicated by higher tensile strength [51], and it mostly depends on the stability of the fibers and the rise in collagen concentration. Therefore, the reason for this might be due to the crude extract strengthening the incision wound by raising collagen levels, which may have caused the wound edges to cling together at the repaired site [52]. This result is consistent with another study conducted on the plan *Rumex abyssinicus*, leave extracts have shown high breaking strength in incision wounds with enhanced fibroblast proliferation, angiogenesis, keratinization, and incised collagen density as compared to the control group [31].

In the current investigation, a partial thickness burn wound model was used to assess the extract's effectiveness in wound healing. Which it show a significant contraction. The process

of producing partial thickness Burn wounds generate three zones: an zone of inflammation immediately adjacent to a zone of hypo perfusion, a zone of hypo perfusion encircling the zone of necrosis and a zone of necrosis at the site of application,. A partial thickness wound becomes a full thickness wound due to microbial colonization, chronic hypo perfusion, excessive free radical generation, and severe inflammation, which complicates and slows the healing process. [53]. In our study, the leaves of *C. myricoides* extract showed significantly increased wound contraction and a reduction in the period of epithelialization. This might be due to the composition of secondary metabolites, and the antibacterial and anti-inflammatory activity of the extract.

Additionally, mice treated with 10% (w/w) of the crude extract and the solvent fraction ointment formulations showed better wound healing effects in terms of increased wound contraction, a shortened epithelialization period, and increased wound tensile strength. Thus, the wound healing effect of *C. myricoides* seems dose-dependent.

For a wound to heal, inflammation is crucial by eliminating debris, germs, and necrotic tissue, as well as by activating and enlisting fibroblasts from an acute wound. In healthy people, inflammation is a self-restraining process. But excessive inflammation hinders the healing of wounds [54]. Inducing hind paw edema with carrageenan is frequently used to test and develop anti-inflammatory medications. In a short period of time, carrageenan injection causes inflammation. It causes three separate periods of inflammation: the first (0–1.5 h), second (1.5–2.5 h), and third (2.5–5 h) are the three phases of inflammation. Histamine, serotonin, bradykinin, and prostaglandins are inflammatory mediators. Since the third phase is susceptible to the majority of therapeutically effective anti-inflammatory medications, this phase is mostly utilized to evaluate the anti-inflammatory effects of traditional medicinal herbs [55]. This investigation showed that an 80% methanol extract of *C. myricoides* had no significant anti-inflammatory effect at a lower dose (100 mg/kg) of the extract. This might be due to inadequate concentration of active constituents in lower doses of extract. Whereas, the higher dose of the extract (200 mg/kg and 400 mg/kg) of *C. myricoides* leaves shows significant (p<0.001) inhibited paw edema as compared to negative control and 100mg/kg extract. The observed edema inhibition was pronounced in the later stages of inflammation, and it was comparable to the effects of non-steroidal anti-inflammatory medications like indomethacin [56]. The finding of this study were in line with that of the methanol extract of *C. petasites* that had shown significant dose-dependent edema reduction in the carrageenan induced animal model [57].

# 6. Conclusion

The present study showed that an 80% methanol extract of the leaves of *C. myricoides*and solvent fractions generally has a wound-healing and anti-inflammatory effect. The aqueous portion of the solvent fractions shows better wound healing activity. The findings of this study offer scientific evidence supporting the traditional claim of *C. myricoides* leaves as a remedy for the treatment of wounds in Ethiopia.

# Supporting information

**S1 File.**
(XLSX)

**S2 File. Author checklist.**
(DOCX)

## Acknowledgments

The authors would like to thank University of Gondar.

## Author Contributions

**Conceptualization:** Alemante Tafese Beyna, Assefa Kebad Mengesha, Ermias Teklehaimanot Yefter, Wubayehu Kahaliw.

**Data curation:** Alemante Tafese Beyna, Wubayehu Kahaliw.

**Formal analysis:** Alemante Tafese Beyna, Ermias Teklehaimanot Yefter, Wubayehu Kahaliw.

**Investigation:** Alemante Tafese Beyna, Wubayehu Kahaliw.

**Methodology:** Alemante Tafese Beyna, Assefa Kebad Mengesha.

**Project administration:** Alemante Tafese Beyna.

**Resources:** Alemante Tafese Beyna, Wubayehu Kahaliw.

**Software:** Alemante Tafese Beyna, Assefa Kebad Mengesha.

**Supervision:** Assefa Kebad Mengesha, Ermias Teklehaimanot Yefter, Wubayehu Kahaliw.

**Validation:** Alemante Tafese Beyna, Assefa Kebad Mengesha.

**Visualization:** Ermias Teklehaimanot Yefter, Wubayehu Kahaliw.

**Writing – original draft:** Alemante Tafese Beyna, Wubayehu Kahaliw.

**Writing – review & editing:** Alemante Tafese Beyna, Assefa Kebad Mengesha, Ermias Teklehaimanot Yefter, Wubayehu Kahaliw.

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
