## [Decision Letter · Decision Letter 0]

18 Apr 2024

PONE-D-24-09941Evaluation of Wound Healing and Anti-Inflammatory Activity of Hydro-alcoholic Extract and Solvent Fractions of the Leaves of Clerodendrum myricoides (Lamiaceae) in MicePLOS ONE

Dear Dr. Beyna,

Thank you for submitting your manuscript to PLOS ONE. After careful consideration, we feel that it has merit but does not fully meet PLOS ONE’s publication criteria as it currently stands. Therefore, we invite you to submit a revised version of the manuscript that addresses the points raised during the review process.

We look forward to receiving your revised manuscript.

Kind regards,

Ahmed E. Abdel Moneim

Academic Editor

PLOS ONE

Journal Requirements:

2. As part of your revision, please complete and submit a copy of the Full ARRIVE 2.0 Guidelines checklist, a document that aims to improve experimental reporting and reproducibility of animal studies for purposes of post-publication data analysis and reproducibility: https://arriveguidelines.org/sites/arrive/files/documents/Author%20Checklist%20-%20Full.pdf Please include your completed checklist as a Supporting Information file. Note that if your paper is accepted for publication, this checklist will be published as part of your article.

3. In the online submission form, you indicated that [Almost all the materials and data of our study are included in the manuscript, a few of the material and data will be available to other researchers upon request]. 

Reviewers' comments:

Reviewer's Responses to Questions

**Comments to the Author**

1. Is the manuscript technically sound, and do the data support the conclusions?

Reviewer #1: Yes

Reviewer #2: Yes

Reviewer #3: Yes

2. Has the statistical analysis been performed appropriately and rigorously? 

Reviewer #1: Yes

Reviewer #2: Yes

Reviewer #3: Yes

3. Have the authors made all data underlying the findings in their manuscript fully available?

Reviewer #1: Yes

Reviewer #2: Yes

Reviewer #3: Yes

4. Is the manuscript presented in an intelligible fashion and written in standard English?

Reviewer #1: Yes

Reviewer #2: Yes

Reviewer #3: No

5. Review Comments to the Author

Reviewer #1: The paper is well organized, well written and ready for direct publication by the journal from my point of view. The flow of thoughts and language are in the optimum level and professional enough for direct consideration as an exceptional article. The author wrote the literature successfully. The conclusion is compatible with results. My recommendation is to "accept" this article

Reviewer #2: Check the bibliographic entries throughout the text, some of which are non-compliant, review some entries in the bibliographic references

lack of The chemical test performed only revealed the qualitative and quantitative assessment of the different extracts, which is not sufficient to justify the motive for selecting different solvent fractions of the test extract. More over, all the different fractions need quantitative assessment of some potential bioactives, like total polyphenolic and flavonoid content determination.

Time or seasonality of plant material collection should be added.

Role medicinal plants should be discussed in the case of drug-resistant bacteria.

Check the bibliographic entries throughout the text, some of which are non-compliant, review some entries in the bibliographic references. some references are too old

Reviewer #3: The manuscript entitled “Evaluation of Wound Healing and Anti-Inflammatory Activity of Hydro-alcoholic Extract and Solvent Fractions of the Leaves of Clerodendrum myricoides (Lamiaceae) in Mice”, introduces new findings but requires significant revisions that undermine its originality.

6. PLOS authors have the option to publish the peer review history of their article (what does this mean?). If published, this will include your full peer review and any attached files.

Reviewer #1: **Yes: **Tafere MULAW Belete

Reviewer #2: **Yes: **Teklie Mengie Ayele

Reviewer #3: No

---

## [Author Response · Author response to Decision Letter 0]

13 May 2024

I wanted to extend my heartfelt appreciation for your diligent review of my manuscript. Your expertise and insightful feedback have been invaluable in shaping the development of this work.

I have carefully reviewed the reviewer's comments and suggestions, and I am grateful for the thoroughness with which they were provided. Your constructive criticism has guided me in refining the manuscript to ensure its accuracy and clarity.

Attached is the revised version of the manuscript, which incorporates the changes, recommended by the reviewer. I trust that these revisions have addressed the concerns raised and have strengthened the overall quality of the manuscript.

reviewer 2.

1. as you mention we include the season of plant collection in the updated manuscript

2. about qualitative biochemical test. To perform qualitative biochemical test our lab set up is limited us. that is the reason why we are not perform. but we will to do in the future .

---

## [Decision Letter · Decision Letter 1]

24 Jun 2024

Evaluation of Wound Healing and Anti-Inflammatory Activity of Hydro-alcoholic Extract and Solvent Fractions of the Leaves of Clerodendrum myricoides (Lamiaceae) in Mice

PONE-D-24-09941R1

Dear Dr. Beyna,

We’re pleased to inform you that your manuscript has been judged scientifically suitable for publication and will be formally accepted for publication once it meets all outstanding technical requirements.

Kind regards,

Ahmed E. Abdel Moneim

Academic Editor

PLOS ONE

Additional Editor Comments (optional):

Reviewers' comments:

Reviewer's Responses to Questions

**Comments to the Author**

1. If the authors have adequately addressed your comments raised in a previous round of review and you feel that this manuscript is now acceptable for publication, you may indicate that here to bypass the “Comments to the Author” section, enter your conflict of interest statement in the “Confidential to Editor” section, and submit your "Accept" recommendation.

Reviewer #1: All comments have been addressed

Reviewer #2: All comments have been addressed

Reviewer #3: All comments have been addressed

2. Is the manuscript technically sound, and do the data support the conclusions?

Reviewer #1: Yes

Reviewer #2: Yes

Reviewer #3: Yes

3. Has the statistical analysis been performed appropriately and rigorously? 

Reviewer #1: Yes

Reviewer #2: Yes

Reviewer #3: Yes

4. Have the authors made all data underlying the findings in their manuscript fully available?

Reviewer #1: Yes

Reviewer #2: Yes

Reviewer #3: Yes

5. Is the manuscript presented in an intelligible fashion and written in standard English?

Reviewer #1: Yes

Reviewer #2: Yes

Reviewer #3: Yes

6. Review Comments to the Author

Reviewer #1: I appreciate the changes in the manuscript that is well adressed the reviewer comment and the article is well written acceptable for piblication.

Reviewer #2: (No Response)

Reviewer #3: The authors made all data underlying the findings in their manuscript fully available and the manuscript presented in an intelligible fashion and written in standard English. The authors have been answered all the questions in a proper way and that all responses meet formatting specifications.

7. PLOS authors have the option to publish the peer review history of their article (what does this mean?). If published, this will include your full peer review and any attached files.

Reviewer #1: **Yes: **Tafere Mulaw Belete

Reviewer #2: **Yes: **Teklie Mengie Ayele

Reviewer #3: No

---

## [Editor Report · Acceptance letter]

1 Jul 2024

PONE-D-24-09941R1 

PLOS ONE

Dear Dr. Beyna, 

I'm pleased to inform you that your manuscript has been deemed suitable for publication in PLOS ONE. Congratulations! Your manuscript is now being handed over to our production team.

Kind regards, 

on behalf of

Dr. Ahmed E. Abdel Moneim 

Academic Editor

PLOS ONE